# Expression Patterns of PAK4 and PHF8 Are Associated with the Survival of Gallbladder Carcinoma Patients

**DOI:** 10.3390/diagnostics13061149

**Published:** 2023-03-17

**Authors:** Ae Ri Ahn, Maryam Karamikheirabad, Min Su Park, Junyue Zhang, Hyun Sun Kim, Ji Su Jeong, Kyoung Min Kim, Ho Sung Park, Kyu Yun Jang

**Affiliations:** 1Department of Pathology, Jeonbuk National University Medical School, Jeonju 54896, Republic of Korea; 2Department of Medicine, Jeonbuk National University Medical School, Jeonju 54896, Republic of Korea; 3Research Institute of Clinical Medicine, Jeonbuk National University, Jeonju 54896, Republic of Korea; 4Research Institute, Jeonbuk National University Hospital, Jeonju 54896, Republic of Korea

**Keywords:** gallbladder, carcinoma, immunohistochemistry, PAK4, PHF8, prognosis

## Abstract

Background: PAK4 and PHF8 are involved in cancer progression and are under evaluation as targets for cancer therapy. However, despite extensive studies in human cancers, there are limited reports on the roles of PAK4 and PHF8 in gallbladder cancers. Methods: Immunohistochemical expression of PAK4 and PHF8 and their prognostic significance were evaluated in 148 human gallbladder carcinomas. Results: PAK4 expression was significantly associated with PHF8 expression in gallbladder carcinomas. Positive expression of nuclear PAK4, cytoplasmic PAK4, nuclear PHF8, and cytoplasmic PHF8 were significantly associated with shorter overall survival and relapse-free survival in univariate analysis. Multivariate analysis showed that nuclear PAK4 expression and nuclear PHF8 expression were independent predictors of overall survival and relapse-free survival in gallbladder carcinomas. Furthermore, coexpression of nuclear PAK4 and nuclear PHF8 predicted shorter overall survival (*p <* 0.001) and relapse-free survival (*p <* 0.001) of gallbladder carcinoma in multivariate analysis. Conclusions: This study suggests that the individual and coexpression patterns of PAK4 and PHF8 as the prognostic indicators for gallbladder carcinoma patients.

## 1. Introduction

P21-activated kinase 4 (PAK4) is a serine/threonine kinase that participates in regulating cell signaling pathways, including tumorigenesis [1,2]. The expression of PAK4 has been shown to be involved in cancer progression [2,3,4]. PAK4 expression was higher in cancer tissue and a higher expression was related to a more aggressive phenotype and a poorer prognosis in various human cancers [5,6,7,8,9]. The mechanism of PAK4 in cancer progression has been suggested by its role in regulating various aspects of cancer cell behavior, such as proliferation, invasiveness, angiogenesis, metabolic reprogramming, and immune evasion [3,4]. In addition, inhibition of PAK4 was presented as a therapeutic strategy for human cancers [3,10] and PAK4 inhibitors showed antitumor activity in the breast [11], pancreas [12], and colorectal cancers [13].

Plant homeodomain finger protein 8 (PHF8) is a histone demethylase [14] and shown to be involved in cell growth, survival, apoptosis, and the epithelial to mesenchymal transition (EMT) of cancer cells [15,16,17,18]. In cancer cells, PHF8 acts as a demethylase removing methyl groups from histone proteins and, thus, altering the expression of genes involved in tumorigenesis [19] and acting as a transcriptional coactivator of oncogenes [20]. Furthermore, a higher expression of PHF8 was observed in various human cancers [15,16,17,21,22] and it was associated with poor clinical outcomes of the gastric [15], liver [17,18], and colorectal cancers [16]. Additionally, inhibition of PHF8 inhibited tumor growth [15,18,20,23]. Therefore, PHF8 has been suggested as a tumor marker and therapeutic target for human cancer [18,20].

Gallbladder cancer is an uncommon cancer which has a relatively high mortality rate [24]. Therefore, understanding the underlying mechanisms of gallbladder cancer progression is crucial in improving the clinical outcomes of patients [25]. The importance of PAK4 and PHF8 in cancer progression and as therapeutic targets has been emphasized in various human cancers, and the inhibition of PAK4 and PHF8 is under evaluation in cancer therapy [3,10,18,20]. Although it has been reported that PAK4 is overexpressed in gallbladder cancer during the evaluation of gene expression profiles of gallbladder cancers [26], studies on PAK4 and PHF8 in gallbladder cancer have been limited. In addition, although it is difficult to find public data for gallbladder cancers, PAK4 expression was significantly associated with PHF8 expression (Pearson’s *r* = 0.31, *p <* 0.001) in human cancers based on the GEPIA database (http://gepia.cancer-pku.cn (accessed on 10 January 2023)) [27]. Moreover, considering that PAK4 shows its kinase activity upon interaction with CDC42 [1] and CDC42 signaling is regulated by PHF8 [23], there might be potential relationships between the roles of PAK4 and PHF8 in the progression of gallbladder cancers. Therefore, this study evaluated the expression patterns and the association of PAK4 and PHF8 in human gallbladder carcinomas (GBCs) and assessed the prognostic significance of their expression patterns.

## 2. Materials and Methods

### 2.1. GBC Patients

This study evaluated patients who underwent surgery for GBC between the years 2000 and 2010. A total of 148 cases with the available medical records, histologic slides, and paraffin-embedded tissue blocks were included in the study. To obtain clinicopathological information, we reviewed the medical records and slides. Thirty-six patients received postoperative chemotherapy, nine patients received radiotherapy, and eight patients received both. No patients received neoadjuvant chemotherapy. Tumor stage and histopathologic factors were reviewed according to the 5th edition of the WHO classification of digestive system tumors [28] and the 8th edition of the American Joint Committee Cancer Staging System [29]. This study was approved by the institutional review board of Jeonbuk National University Hospital (IRB number CUH 2023-01-002).

### 2.2. Immunohistochemical Staining and Scoring

Immunohistochemical staining for PAK4 and PHF8 in GBCs was performed by using tissue microarray (TMA) sections containing two 3.0 mm cores per case. The TMA tissue sections underwent deparaffinization and antigen retrieval by boiling in a pH 6.0 antigen retrieval solution (DAKO, Glostrup, Denmark) for 20 min using a microwave oven. The tissue sections were incubated with primary antibodies for PAK4 (Cat# 14685-1-AP, 1:100, Proteintech, Chicago, IL, USA) and PHF8 (Cat# IHC-00343, 1:100, Bethyl Laboratories, Montgomery, TX, USA) and then visualized using the DAKO Envision system (DAKO, Carpinteria, CA, USA). Slides that underwent immune staining of PAK4 and PHF8 were evaluated based on the intensity and extent of the staining in the cytoplasm or nuclei of tumor cells [30,31,32]. The staining intensity and area were rated on scales of 0 to 3 (0; no staining, 1; weak, 2; intermediate, 3; strong) and 0 to 5 (0; 0%, 1; 1%, 2; 2–10%, 3; 11–33%, 4; 34–66%, 5; 67–100%), respectively [30,31,32]. The final immunohistochemical score was obtained by adding the staining intensity and area scores for each of the two TMA cores, resulting in a score ranging from zero to sixteen. The slides were evaluated by two pathologists (K.Y.J. and H.S.P.) who reached a consensus score through simultaneous observation. The entire process was performed blind to the clinicopathological information.

### 2.3. Statistical Analysis

PAK4 and PHF8 expression positivity was assessed using receiver operating characteristic (ROC) curve analysis [33] to predict patient mortality through December 2014. Survival analysis performed for overall survival (OS) and relapse-free survival (RFS). An OS event was defined as the death of a patient due to GBC and the duration of follow-up was calculated from the date of the operation to the date of last contact or death. The patients who were either alive or had died from other causes at the end of the follow-up were censored. An RFS event was defined as a relapse of GBC or death from GBC, and the duration of follow-up was calculated from the date of operation to the date of the event or last contact. The patients who were either alive without recurrence or had died from other causes at the end of the follow-up were censored. Statistical analysis was performed using SPSS software (version 22.0, IBM, Armonk, NY, USA) for univariate and multivariate Cox proportional hazards regression analysis, Kaplan–Meier survival analysis, and Pearson’s chi-square test. A *p*-value of less than 0.05 was considered statistically significant.

## 3. Results

### 3.1. The Expression of PAK4 and PHF8 in GBCs

Immunohistochemical expression of PAK4 and PHF8 was observed in both the nuclei and cytoplasm of tumor cells in human GBC tissue (Figure 1A). The subcellular distribution of PAK4 and PHF8 expression in carcinoma tissue from human gallbladders differed between cases, with some cases having expression primarily in the cytoplasm, others in the nuclei, and some in both the nuclei and cytoplasm of tumor cells (Figure 1A). As a result, PAK4 and PHF8 expression were evaluated separately based on nuclear and cytoplasmic localization: nuclear expression of PAK (n-PAK4), cytoplasmic expression of PAK4 (c-PAK4), nuclear expression of PHF8 (n-PHF8), and cytoplasmic expression of PHF8 (c-PHF8). Positivity for the immunohistochemical expression of n-PAK4, c-PAK4, n-PHF8, and c-PHF8 were determined by evaluating cut-off points through ROC curve analysis (Figure 1B). The cut-off points for n-PAK4, c-PAK4, n-PHF8, and c-PHF8 were 12, 15, 14, and 6, respectively (Figure 1B). Positivity for n-PAK4 was significantly associated with the T category of tumor stage, higher histologic grade, and the expression of c-PAK4, n-PHF8, and c-PHF8 (Table 1). Positivity for c-PAK4 showed a significant association with the T category of tumor stage, higher histologic grade, and the expression of n-PHF8, and c-PHF8 (Table 1). n-PHF8 positivity was significantly associated with the T category of tumor stage, higher histologic grade, and the expression of c-PHF8 (Table 1). Positivity for c-PHF8 showed a significant association with the age of the patients and the T category of tumor stage (Table 1).

### 3.2. The Expression Patterns of PAK4 and PHF8 Are Associated with Shorter Survival of GBC Patients

The significant factors associated with OS or RFS in univariate analysis were age, serum level of CA19-9, tumor stage, T category of the stage, lymph node metastasis, distant metastasis, histologic type, histologic grade, and the expression of n-PAK4, c-PAK4, n-PHF8, and c-PHF8 (Table 2). Patients with n-PAK4 positivity have a 3.063 times higher risk of death [95% CI (95% confidential interval), 1.972–4.759; *p <* 0.001] and a 2.798 times higher risk of relapse or death (95% CI, 1.835–4.268; *p <* 0.001) (Table 2). Similarly, c-PAK4 positivity results indicate a 1.828 times higher risk of death (95% CI, 1.194–2.797; *p* = 0.005) and a 1.651 times higher risk of relapse or death (95% CI, 1.095–2.489; *p* = 0.017) (Table 2). The expression of n-PHF8 also increases the risk of death by 3.713 times (95% CI, 2.378–5.799; *p <* 0.001) and the risk of relapse or death by 3.471 times (95% CI, 2.265–5.319; *p <* 0.001) (Table 2). c-PHF8 positivity leads to a 2.240 times higher risk of death (95% CI, 1.397–3.252; *p <* 0.001) and a 1.941 times higher risk of relapse or death (95% CI, 1.243–3.030; *p* = 0.004) (Table 2). The Kaplan–Meier survival curves are presented in Figure 2, demonstrating the impact of n-PAK4, c-PAK4, n-PHF8, and c-PHF8 expression on the survival of GBC patients.

Multivariate survival analysis was conducted incorporating the following factors that showed significance in univariate analysis: age, CA19–9 level, tumor stage, T category of the stage, lymph node metastasis, distant metastasis, lymphovascular invasion, histologic type, histologic grade, and the expression of n-PAK4, c-PAK4, n-PHF8, and c-PHF8. Multivariate analysis indicated that the age, tumor stage, the T category of the stage, n-PAK4 expression, and n-PHF8 expression were independent predictors of OS and RFS in GBC patients (Table 3). The patients with positive n-PAK4 expression had a 2.003-fold higher risk of OS (95% CI, 1.161–3.458; *p =* 0.013) and a 1.794-fold higher risk of RFS (95% CI, 1.082–2.975; *p =* 0.024) compared to those with negative n-PAK4 expression. Similarly, the patients with positive n-PHF8 expression had a 2.130-fold higher risk of OS (95% CI, 1.229–3.695; *p =* 0.007) and a 2.411-fold higher risk of RFS (95% CI, 1.408–4.127; *p =* 0.001) compared to those with negative n-PHF8 expression (Table 3).

### 3.3. Coexpression Patterns of Nuclear PAK4 and Nuclear PHF8 Predict Survival of GBC Patients

The expression of n-PAK4 and n-PHF8 were independent indicators of OS and RFS for GBC patients. In addition, there was a significant association between n-PAK4 and n-PHF8, as demonstrated in Table 1. Therefore, we further evaluated the prognostic significance of the coexpression patterns of n-PAK4 and n-PHF8 in GBCs. The GBCs were initially divided into four subgroups based on the coexpression patterns of n-PAK4 and n-PHF8: n-PAK4^−^/n-PHF8^−^, n-PAK4^+^/n-PHF8^−^, n-PAK4^−^/n-PHF8^+^, and n-PAK4^+^/n-PHF8^+^. The n-PAK4^−^/n-PHF8^−^ subgroup had the longest OS (10-year OS, 70%) and RFS (10-year RFS, 64%), while the n-PAK4^+^/n-PHF8^+^ subgroup had the shortest OS (10-year OS, 7%) and RFS (10-year RFS, 7%) (Table 4) (Figure 3A). However, there was no significant difference in OS and RFS between the n-PAK4^+^/n-PHF8^-^ subgroup and the n-PAK4^−^/n-PHF8^+^ subgroup (Figure 3A). Therefore, the GBCs were regrouped into three prognostic subgroups: [n-PAK4^−^/n-PHF8^−^], [n-PAK4^+^/n-PHF8^−^ or n-PAK4^−^/n-PHF8^+^], and [n-PAK4^+^/n-PHF8^+^]. This subgrouping of GBCs based on the coexpression patterns of n-PAK4 and n-PHF8 into three subgroups was significantly associated with OS and RFS (multivariate analysis: OS, overall *p <* 0.001; RFS, overall *p <* 0.001) (Table 5) (Figure 3B).

### 3.4. Expression Patterns of Nuclear PAK4 and Nuclear PHF8 Are Associated with Survival of GBC Patients Who Received Postoperative Therapies

In 37 patients who received postoperative chemotherapy and/or radiotherapy, individual expression of n-PAK4 and n-PHF8 were significantly associated with OS and RFS of GBC patients (Figure 4A). In addition, coexpression patterns of n-PAK4 and n-PHF8 were also significantly associated with OS and RFS of GBC patients (Figure 4B).

## 4. Discussion

In this study, the expression of PAK4 and PHF8 were significantly associated with each other and their expression patterns predicted the shorter survival of GBC patients. Both n-PAK4 and n-PHF8 expression were independent indicators of shorter OS and RFS of GBCs in multivariate analysis. Consistently, a higher expression of PAK4 was associated with poor prognosis in various cancers, including breast [5,6], lung [9], ovarian [8], and gastric cancer [7]. In breast cancer, PAK4 expressed higher in cancer tissue than in normal tissue [5]. Moreover, elevated PAK4 mRNA expression in cancer tissue was strongly correlated with shorter disease-specific survival in breast cancer patients [5]. Higher PAK4 expression was significantly related to advanced tumor stage, lymph node metastasis, and shorter survival of gastric carcinoma patients [7]. In lung cancer, a higher expression of the PAK4 gene, as indicated by microarray analysis, predicted shorter OS of non-small cell lung cancers, but the immunohistochemical expression of PAK4 did not predict the OS of patients [9]. Similarly, in the prostate, the expression of PHF8 was higher in cancer tissues than their normal counterparts [21]. In addition, the expression of PHF8 was upregulated in gastric cancer and the overexpression of PHF8 predicted shorter OS of HER2-negative gastric cancers [22]. Higher expression of PHF8 has also been observed in various human cancers and has been associated with shorter survival of liver cancer [17,18], colorectal cancer [16], gastric cancer [15], and lung cancer patients [23]. These results suggest that the expression of PAK4 and PHF8 might be used as markers to predict the survival of human cancers including GBCs.

In GBCs, the expression of PAK4 and PHF8 was observed in both the nuclei and cytoplasm of tumor cells. Therefore, we evaluated the expression patterns of PAK4 and PHF8 based on their nuclear and cytoplasmic expression and found that the nuclear expression patterns of PAK4 and PHF8 were more predictive of survival for GBC patients than was their cytoplasmic expression. Similarly, there are reports indicating that PAK4 and PHF8 are expressed in both the nuclei and cytoplasm of cancer cells [6,8,21]. In breast cancer, PAK4 was expressed in both the nuclei and cytoplasm of cancer cells, and a higher expression of PAK4 predicted the shorter survival of patients [6]. In ovarian cancers, a higher expression of both n-PAK4 and c-PAK4 was significantly associated with shorter OS and disease-free survival in univariate analysis [8]. However, in multivariate analysis, only c-PAK4 expression was an independent indicator of shorter OS in patients with ovarian carcinoma [8]. In gastric carcinoma tissue, PAK4 was mainly expressed in the cytoplasm of tumor cells and a higher expression of c-PAK4 predicted shorter survival for gastric carcinoma patients [7]. The immunohistochemical expression of PAK4 in lung cancer was mainly observed in the cytoplasm of tumor cells, but a higher expression of c-PAK4 did not predict OS for patients [9]. In addition, there are various reports on the subcellular localization of PHF8 and its prognostic significance in cancer patients. In hepatocellular carcinomas, PHF8 expression was observed in nuclei, and a higher expression of n-PHF8 predicted the shorter survival of patients [18]. In contrast, a higher expression of PHF8 in gastric cancer tissue than in normal tissue was mainly observed in the cytoplasm of cancer cells [15]. In prostate cancer, PHF8 was expressed in both the nuclei and cytoplasm of tumor cells [21]. Therefore, subcellular localization of the expression of PAK4 and PHF8 might be different according to the type of cancer and biological status of cancer cells. However, despite this variable expression pattern, a higher expression of PAK4 and PHF8 might be used as a prognostic indicator for cancer patients.

The prognostic impact of the expression of PAK4 and PHF8 might be associated with their role in cancer progression. Both PAK4 and PHF8 have roles in the regulation of various cellular processes that are involved in cancer progression, including cell proliferation, survival from apoptosis, migration, angiogenesis, metabolic regulation, and immune evasion of cancer cells [2,4,5,15,16,17,18,34]. A higher expression of PAK4 promoted proliferation and invasiveness through activation of the PI3K/AKT pathway in breast cancer cells [6] and through c-Src and the EGFR pathway in ovarian cancer cells [8]. In breast cancer cells, PAK4 was involved in the progression of cancer by preventing senescence-like growth arrest [5]. PAK4 induced resistance to immune checkpoint inhibitors by changing the tumor microenvironment [34]. A higher expression of PHF8 also stimulated proliferation, metastasis, and the EMT phenotype of cancer cells by activating the Wnt/β-catenin signaling in lung cancer cells [23] and gastric cancer cells [15]. Therefore, based on the oncogenic roles of PAK4 and PHF8, variable strategies to inhibit them have been evaluated in various cancers. Especially, a higher expression of PAK4 was associated with resistance to anticancer therapy [11,12]. A higher expression of PAK4 was associated with the shorter survival of breast cancer patients who are treated with endocrine therapy or tamoxifen [11]. PAK4 was involved in resistance to tamoxifen of breast cancer cells [11] and resistance to gemcitabine of pancreatic cancer cells [12]. Inhibition of PAK4 with specific inhibitors has been shown to inhibit the proliferation of cancer cells [10] and induced restoration of sensitivity to anticancer agents [11,12]. Moreover, as more evidence emerges regarding the role of PAK4 in the immune evasion of cancer cells, research has evaluated the efficacy of a combination treatment of PAK4 inhibition and immune checkpoint inhibitors [34,35]. Higher levels of PAK4 expression were associated with reduced immune cell infiltration and contributed to making melanoma cells resistant to anti-PD-1 therapy through the β-catenin pathway [35]. In addition, PAK4 inhibition enhanced the effectiveness of PD-1 blockade therapy in melanoma cells [34,35]. A higher expression of PHF8 was also associated with resistance to anticancer therapy [20]. PHF8 acted as coactivator to induce HER2 expression in breast cancer cells and contributed in resistance to trastuzumab [20]. Knockdown of PHF8 attenuated HER2 overexpression-induced proliferation of cancer cells [20]. The effects of PHF8 inhibition on the suppression of cancer cells have been reported in gastric [15], lung [23], and colorectal cancer cells [16]. Therefore, when considering the shorter survival of GBC patients, having tumors with an elevated expression of PAK4 and/or PHF8, PAK4, and PHF8 might be potential therapeutic targets for GBC patients.

Another interesting finding of this study is that the coexpression pattern of n-PAK4 and n-PHF8 is very useful in predicting the survival of GBC patients. GBC patients with tumors with an n-PAK4^−^/n-PHF^−^ phenotype had the longest survival time, while those with tumors with an n-PAK4^+^/n-PHF8^+^ phenotype had the shortest survival time. These results suggest a possible cooperative role of PAK4 and PHF8 in cancer progression. When considering the divergent roles of PAK4 and PHF8 in cancer progression, there might be a direct or indirect association between PAK4 and PHF8. The positive correlation between the expression of PAK4 and PHF8 in GBCs suggests that these two molecules might be regulated by common transcription factors, chromatin structure changes, or signaling pathways. Additionally, PHF8 might regulate PAK4 activity and changes in PAK4 activity might, in turn, affect the expression of PHF8. One possibility is that the association between PHF8 and PAK4 might be mediated by CDC42, as there was a report indicating that PHF8 regulates the PKCα-Src-CDC42 signaling cascade [22], and PAK4 was activated when it was bound to active CDC42 [36]. Although the mechanism underlying the association between PHF8 and PAK4 is not clear, the positive correlation between their expression in GBCs implies that these two molecules might be functionally linked. Therefore, further study is needed to determine the exact nature of this relationship and its significance in cancer progression, especially in gallbladder cancer.

## 5. Conclusions

In conclusion, this study investigated the expression of PAK4 and PHF8 in human GBC tissue and its correlation with clinicopathologic variables and patient outcomes. The results showed that both the cytoplasmic and nuclear expression patterns of PAK4 and PHF8 were associated with the shorter survival of GBC patients. Especially, the coexpression patterns of n-PAK4 and n-PHF8 were predictive of the survival of GBC patients. Therefore, this study suggests that the expression of PAK4 and PHF8 might be used to predict the prognosis of GBCs and suggests a potential for PAK4 and PHF8 as targets for novel therapeutic strategies for GBCs.

## Figures and Tables

**Figure 1 diagnostics-13-01149-f001:**
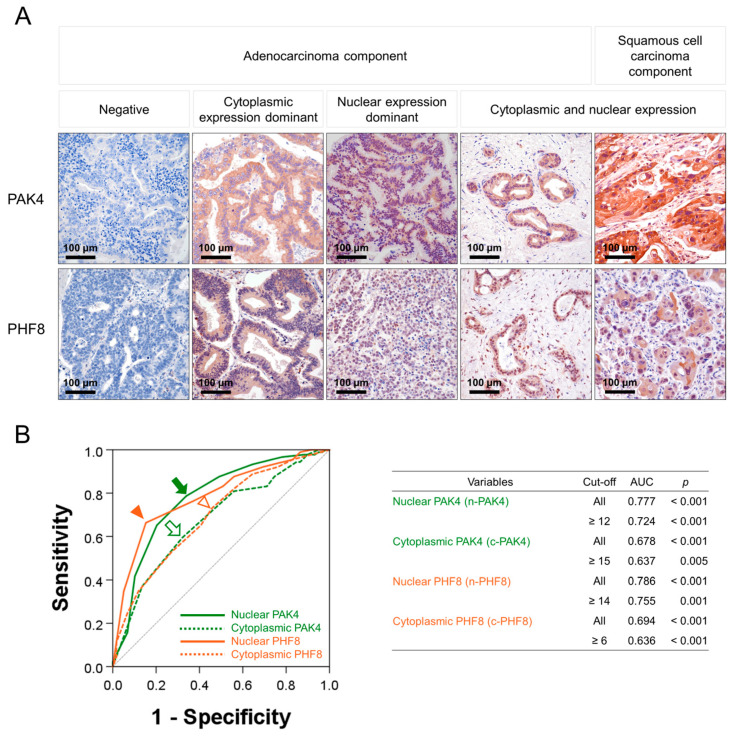
Immunohistochemical expression of PAK4 and PHF8 in human GBCs. (**A**) PAK4 and PHF8 are expressed in the nuclei and cytoplasm of GBC cells. Original magnification: ×400. (**B**) Receiver operating characteristic curve analysis to determine cut-off points of immunohistochemical expression of PAK4 and PHF8. The cut-off point was determined at the point with the highest area under the curve (AUC). Green arrow (nuclear expression of PAK4, n-PAK4), empty green arrow (cytoplasmic expression of PAK4, c-PAK4), orange arrowhead (nuclear expression of PHF8, n-PHF8), and empty green arrowhead (cytoplasmic expression of PHF8, c-PHF8) indicate the cut-off points.

**Figure 2 diagnostics-13-01149-f002:**
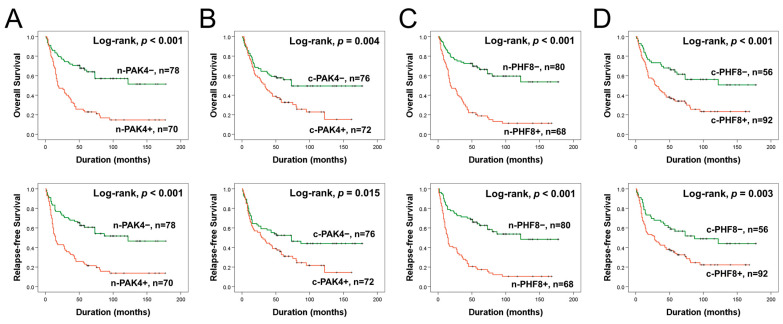
Kaplan–Meier survival curves of OS and RFS according to the expression of nuclear PAK4 (n-PAK4) (**A**), cytoplasmic PAK4 (c-PAK4) (**B**), nuclear PHF8 (n-PHF8) (**C**), and cytoplasmic PHF8 (c-PHF8) (**D**) in GBCs.

**Figure 3 diagnostics-13-01149-f003:**
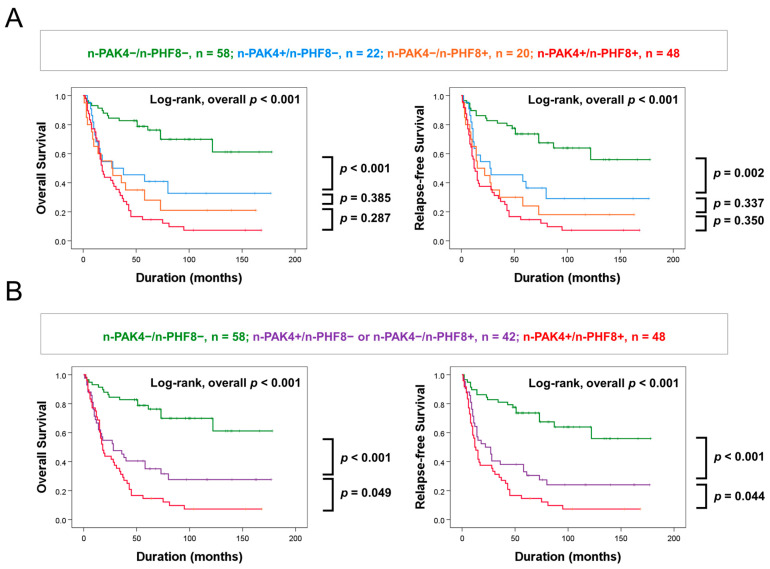
Kaplan–Meier survival analysis according to the combined expression pattern of n-PAK4 and n-PHF8. (**A**) Kaplan–Meier survival curves in four prognostic subgroups of GBCs according to the combined expression pattern of n-PAK4 and n-PHF8: [n-PAK4^−^/n-PHF8^−^], [n-PAK4^+^/n-PHF8^−^], [n-PAK4^−^/n-PHF8^+^], and [n-PAK4^+^/n-PHF8^+^] subgroups. (**B**) Survival analysis in three prognostic subgroups of GBCs: [n-PAK4^−^/n-PHF8^−^], [n-PAK4^+^/n-PHF8^-^ or n-PAK4^−^/n-PHF8^+^], and [n- = PAK4^+^/n-PHF8^+^] subgroups.

**Figure 4 diagnostics-13-01149-f004:**
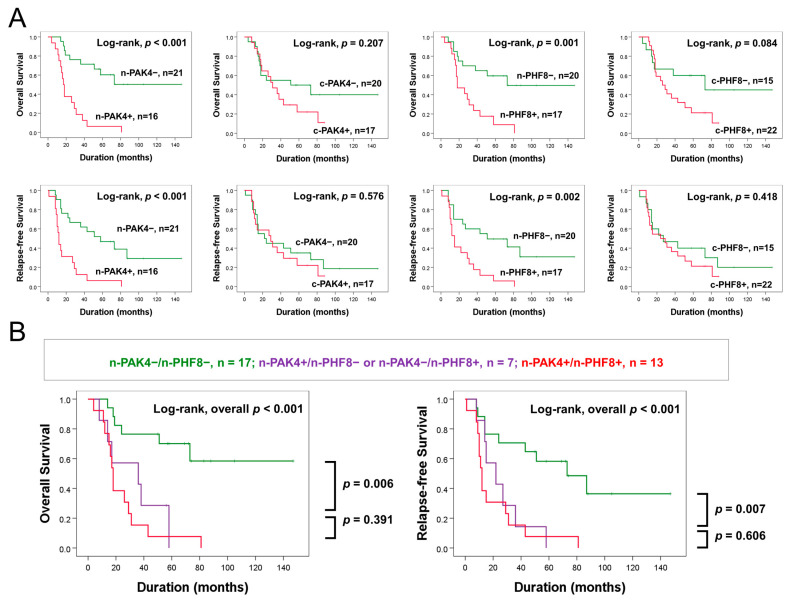
Survival analysis according to the individual and combined expression patterns of n-PAK4 and n-PHF8 in 37 patients who received postoperative therapy. (**A**) Kaplan–Meier survival curves according to the expression of n-PAK4, c-PAK4, n-PHF8, and c-PHF8. (**B**) Survival analysis in three prognostic subgroups of GBCs: [n-PAK4^−^/n-PHF8^−^], [n-PAK4^+^/n-PHF8^−^ or n-PAK4^−^/n-PHF8^+^] and [n-PAK4^+^/n-PHF8^+^] subgroups.

**Table 1 diagnostics-13-01149-t001:** The expression of PAK4 and PHF8 in GBCs.

Characteristics		No.	n-PAK4		c-PAK4		n-PHF8		c-PHF8	
			Positive	*p*	Positive	*p*	Positive	*p*	Positive	*p*
Age (years)	<65 y	69	30 (43%)	0.384	30 (43%)	0.240	28 (41%)	0.221	36 (52%)	0.019
	≥65 y	79	40 (51%)		42 (53%)		40 (51%)		56 (71%)	
Sex	Male	73	32 (44%)	0.405	37 (51%)	0.625	28 (38%)	0.068	45 (62%)	0.898
	Female	75	38 (51%)		35 (47%)		40 (53%)		47 (63%)	
CEA *	≤5.2 ng/mL	119	57 (48%)	0.853	58 (49%)	0.910	54 (45%)	0.431	74 (62%)	0.678
	>5.2 ng/mL	24	11 (46%)		12 (50%)		13 (54%)		16 (67%)	
CA19-9 **	≤37 U/mL	100	44 (44%)	0.297	46 (46%)	0.282	47 (47%)	0.957	64 (64%)	0.356
	>37 U/mL	43	23 (53%)		24 (56%)		20 (47%)		24 (56%)	
Tumor stage	I and II	92	39 (42%)	0.125	43 (47%)	0.551	37 (40%)	0.073	56 961%)	0.678
	III and IV	56	31 (55%)		29 (52%)		31 (55%)		36 (64%)	
T category	T1	41	11 (27%)	0.004	12 (29%)	0.033	14 (34%)	0.010	18 (44%)	0.018
	T2	73	36 (49%)		42 (58%)		30 (41%)		51 (70%)	
	T3	30	21 (70%)		16 (53%)		21 (70%)		19 (63%)	
	T4	4	2 (50%)		2 (50%)		3 (75%)		4 (100%)	
Lymph node metastasis	Absence	111	52 (47%)	0.849	53 (48%)	0.704	51 (46%)	1.000	70 (63%)	0.695
	Presence	37	18 (49%)		19 (51%)		17 (46%)		22 (59%)	
Distant metastasis	Absence	140	64 (46%)	0.107	67 (48%)	0.420	63 (45%)	0.334	85 (61%)	0.129
	Presence	8	6 975%)		5 (63%)		5 (63%)		7 (88%)	
Lymphovascular invasion	Absence	127	60 (47%)	0.975	60 (47%)	0.401	62 (49%)	0.085	79 (62%)	0.979
	Presence	21	10 (48%)		12 (57%)		6 (29%)		13 (62%)	
Histologic type	Adenocarcinoma	143	67 947%)	0.566	68 (48%)	0.327	65 (45%)	0.544	87 (61%)	0.207
	Adenosquamous carcinoma	4	2 (50%)		3 (75%)		2 (50%)		4 (100%)	
	Squamous cell carcinoma	1	1 (100%)		1 (100%)		1 9100%)		1 (100%)	
Histologic grade	G1	63	21 (33%)	0.003	23 (37%)	0.011	20 (32%)	0.003	35 (56%)	0.154
	G2 and G3	85	49 (58%)		49 (58%)		48 (56%)		57 (67%)	
c-PHF8	Negative	56	20 (36%)	0.028	18 (32%)	0.002	14 (25%)	<0.001		
	Positive	92	50 (54%)		54 (59%)		54 (59%)			
n-PHF8	Negative	80	22 (28%)	<0.001	32 (40%)	0.022				
	Positive	68	48 (71%)		40 (59%)					
c-PAK4	Negative	76	25 (33%)	<0.001						
	Positive	72	45 (63%)							

* Carcinoembryonic antigen (CEA) was not measured in six patients. ** Carbohydrate antigen 19-9 (CA19-9) was not measured in five patients.

**Table 2 diagnostics-13-01149-t002:** Univariate Cox regression analysis in GBCs.

Characteristics	No.	OS		RFS	
		HR (95% CI)	*p*	HR (95% CI)	*p*
Age, y ≥ 65 (vs. <65)	79/148	2.248 (1.444–3.500)	<0.001	2.017 (1.317–3.088)	0.001
Sex, female (vs. male)	75/148	0.704 (0.463–1.070)	0.100	0.708 (0.471–1.063)	0.096
CEA, >5.2 ng/mL (vs. ≤5.2 ng/mL) *	24/142	1.382 (0.812–2.351)	0.233	1.286 (0.758–2.180)	0.351
CA19-9, >37 U/mL (vs. ≤37 U/mL) **	43/143	1.818 (1.177–2.811)	0.007	1.892 (1.239–2.889)	0.003
Tumor stage, III and IV (vs. I and II)	56/148	3.555 (2.321–5.445)	<0.001	3.273 (2.160–4.958)	<0.001
T category, T1	41/148	1	<0.001	1	<0.001
T2	73/148	2.279 (1.241–4.183)	0.008	2.416 (1.345–4.339)	0.003
T3	30/148	10.123 (5.231–19.588)	<0.001	8.504 (4.458–16.223)	<0.001
T4	4/148	9.876 (3.157–30.899)	<0.001	9.694 (3.118–30.138)	<0.001
Lymph node metastasis, presence (vs. absence)	37/148	1.982 (1.266–3.104)	0.003	1.997 (1.288–3.098)	0.002
Distant metastasis, presence (vs. absence)	8/148	6.190 (2.849–13.451)	<0.001	4.781 (2.224–10.278)	<0.001
Lymphovascular invasion, presence (vs. absence)	21/148	2.333 (1.364–3.993)	0.002	2.307 (1.354–3.930)	0.002
Histologic type, adenocarcinoma	143/148	1	0.002	1	0.002
adenosquamous carcinoma	4/148	3.772 (1.366–10.419)	0.010	4.013 (1.451–11.099)	0.007
squamous cell carcinoma	1/148	1.804 (1.779–107.101)	0.012	12.753 (1.654–98.308)	0.015
Histologic grade, G2 and G3 (vs. G1)	85/148	3.066 (1.923–4.890)	<0.001	3.043 (1.938–4.778)	<0.001
n-PAK4, positive (vs. negative)	70/148	3.063 (1.972–4.759)	<0.001	2.798 (1.835–4.268)	<0.001
c-PAK4, positive (vs. negative)	72/148	1.828 (1.194–2.797)	0.005	1.651 (1.095–2.489)	0.017
n-PHF8, positive (vs. negative)	68/148	3.713 (2.378–5.799)	<0.001	3.471 (2.265–5.319)	<0.001
c-PHF8, positive (vs. negative)	92/148	2.240 (1.397–3.252)	<0.001	1.941 (1.243–3.030)	0.004

* CEA was not measured in six patients. ** CA19-9 was not measured in five patients. Abbreviations: HR, hazard ratio; 95% CI, 95% confidence interval.

**Table 3 diagnostics-13-01149-t003:** Multivariate Cox regression analysis in GBCs.

Characteristics	OS		RFS	
	HR (95% CI)	*p*	HR (95% CI)	*p*
Age, y ≥65 (vs. <65)	2.730 (1.710–4.358)	<0.001	2.275 (1.455–3.557)	<0.001
Tumor stage, III and IV (vs. I and II)	2.543 (1.337–4.836)	0.004	2.008 (1.050–3.839)	0.035
T category, T1	1	0.031	1	0.030
T2	1.744 (0.911–3.339)	0.093	1.831 (0.979–3.426)	0.058
T3	3.516 (1.439–8.592)	0.006	3.278 (1.377–7.802)	0.007
T4	5.077 (1.401–18.405)	0.013	5.430 (1.522–19.839)	0.009
n-PAK4, positive (vs. negative)	2.003 (1.161–3.458)	0.013	1.794 (1.082–2.975)	0.024
n-PHF8, positive (vs. negative)	2.130 (1.229–3.695)	0.007	2.411 (1.408–4.127)	0.001

Abbreviations: HR, hazard ratio; 95% CI, 95% confidence interval.

**Table 4 diagnostics-13-01149-t004:** Five- and ten-year survival according to coexpression patterns of n-PAK4 and n-PHF8 in GBCs.

Coexpression Pattern of n-PAK4 and n-PHF8	No.	5-Year OS (%)	10-Year OS (%)	5-Year RFS (%)	10-Year RFS (%)
Coexpression Model 1					
n-PAK4/n-PHF8, −/−	58	79	70	74	64
n-PAK4/n-PHF8, +/−	22	41	33	41	29
n-PAK4/n-PHF8, −/+	20	28	21	24	18
n-PAK4/n-PHF8, +/+	48	15	7	15	7
Coexpression Model 2					
n-PAK4/n-PHF8, −/−	58	79	70	74	64
n-PAK4/n-PHF8, −/+ or +/−	42	35	28	33	24
n-PAK4/n-PHF8, +/+	48	15	7	15	7

**Table 5 diagnostics-13-01149-t005:** Univariate and multivariate Cox regression analysis according to coexpression patterns of n-PAK4 and n-PHF8 in GBCs.

Characteristics	OS		RFS	
	HR (95% CI)	*p*	HR (95% CI)	*p*
Univariate analysis				
n-PAK4/n-PHF8, −/−	1	<0.001	1	<0.001
−/+ or +/−	3.727 (2.021–6.873)	<0.001	3.277 (1.847–5.814)	<0.001
+/+	6.019 (3.369–10.751)	<0.001	5.309 (3.077–9.159)	<0.001
Multivariate analysis *				
Age, y ≥65 (vs. <65)	3.139 (1.944–5.068)	<0.001	2.663 (1.699–4.173)	<0.001
CA19-9, >37 U/mL (vs. ≤37 U/mL)			1.569 (1.005–2.451)	0.048
Tumor stage, III and IV (vs. I and II)	2.812 (1.466–5.397)	0.002	3.373 (2.118–5.373)	<0.001
T category, T1	1	0.030		
T2	1.569 (0.820–3.000)	0.173		
T3	3.283 (1.348–7.996)	0.009		
T4	5.368 (1.478–19.487)	0.011		
Histologic grade, G2 and G3 (vs. G1)			2.164 (1.338–3.500)	0.002
n-PAK4/n-PHF8, −/−	1	<0.001	1	<0.001
−/+ or +/−	4.975 (2.611–9.479)	<0.001	4.225 (2.323–7.687)	<0.001
+/+	5.173 (2.769–9.665)	<0.001	5.127 (2.872–9.152)	<0.001

* Multivariate survival analysis was performed with the factors significantly associated with OS or RFS in univariate analysis. Abbreviations: HR, hazard ratio; 95% CI, 95% confidence interval, n-PAK4/n-PHF8, coexpression pattern of n-PAK4 and n-PHF8.

## Data Availability

The datasets generated during and/or analyzed during the current study are available from the corresponding author on reasonable request.

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
