# Peer review of "Expression Patterns of PAK4 and PHF8 Are Associated with the Survival of Gallbladder Carcinoma Patients"

_diagnostics, 2023, doi:10.3390/diagnostics13061149_

Round 1

Reviewer 1 Report

In this manuscript, authors clearly demonstrated that PAK4 and PHF8 are important for the survival of gallbladder carcinoma patients. The experimental data are presented in a way that is extremely easy for readers to understand, and this manuscript should be widely known. Of course, no additional experiments are necessary. As far as I know, this manuscript is a perfect one with no points that need to be changed.

Author Response

Response to reviewer 1

We thank the reviewer for these insightful comments.

Reviewer reports:

 In this manuscript, authors clearly demonstrated that PAK4 and PHF8 are important for the survival of gallbladder carcinoma patients. The experimental data are presented in a way that is extremely easy for readers to understand, and this manuscript should be widely known. Of course, no additional experiments are necessary. As far as I know, this manuscript is a perfect one with no points that need to be changed.

 We thank the reviewer very much for this comment.

Reviewer 2 Report

The authors investigated the expression of PAK4 and PHF8 proteins in gallbladder carcinoma. The idea is new, the methods are appropriate and adequate, the results are interesting and well-presented, and the discussion sounds meaningful. However, a few points remain to be addressed:  

-Line 167, Table 1: the title of Table 1 needs correction.

-Please include the definitions for elevated (vs. normal) levels of CEA and CD19-9.

-Please include how the histologic grade was categorized into low- and high-grade groups. Tumor grade is usually reported on a three-level scale, from G1 to G3.   

-It is important to include the postoperation therapy-related variables (treated vs. not treated, chemotherapy vs. radiotherapy vs. both) in the analysis. I wonder whether the prognostic effects of PAK4 and PHF8 are independent of the therapy received by the patient.

Author Response

Response to reviewer 2

We thank the reviewer for these insightful comments.

Reviewer reports:

 The authors investigated the expression of PAK4 and PHF8 proteins in gallbladder carcinoma. The idea is new, the methods are appropriate and adequate, the results are interesting and well-presented, and the discussion sounds meaningful. However, a few points remain to be addressed: 

 We thank the reviewer for this comment.

-Line 167, Table 1: the title of Table 1 needs correction.

 We thank the reviewer for this comment and apologize for the mistake in preparing the manuscript. In response to the comment of the reviewer, we have revised the title of Table 1.

-Please include the definitions for elevated (vs. normal) levels of CEA and CA19-9.

 We thank the reviewer for this comment. In response to the comment of the reviewer, we have we have included the levels of CEA (5.2 ng/mL) and CA19-9 (37 U/mL) in the Tables.

-Please include how the histologic grade was categorized into low- and high-grade groups. Tumor grade is usually reported on a three-level scale, from G1 to G3.  

 We thank the reviewer for this comment. In response to the comment of the reviewer, we have revised the tables to use the terms 'G1' for low-grade and 'G2 and G3' for high-grade. We hope that this change improves the clarity and accuracy of our manuscript.

-It is important to include the postoperation therapy-related variables (treated vs. not treated, chemotherapy vs. radiotherapy vs. both) in the analysis. I wonder whether the prognostic effects of PAK4 and PHF8 are independent of the therapy received by the patient.

 We thank the reviewer for this comment. In response to the comment of the reviewer, we have included Figure 4, which shows the survival curves for 37 patients who received postoperative chemotherapy and/or radiotherapy based on their individual and combined expression patterns of nuclear PAK4 and nuclear PHF8.